# Extensive Placental Methylation Profiling in Normal Pregnancies

**DOI:** 10.3390/ijms22042136

**Published:** 2021-02-21

**Authors:** Ornella Rondinone, Alessio Murgia, Jole Costanza, Silvia Tabano, Margherita Camanni, Luigi Corsaro, Laura Fontana, Patrizia Colapietro, Luciano Calzari, Silvia Motta, Carlo Santaniello, Tatjana Radaelli, Enrico Ferrazzi, Silvano Bosari, Davide Gentilini, Silvia Maria Sirchia, Monica Miozzo

**Affiliations:** 1Research Laboratories Coordination Unit, Fondazione IRCCS Ca’ Granda Ospedale Maggiore Policlinico, 20122 Milan, Italy; ornella.rondinone@unimi.it (O.R.); alessio.murgia@policlinico.mi.it (A.M.); jole.costanza@policlinico.mi.it (J.C.); margherita.camanni@policlinico.mi.it (M.C.); laura.fontana@unimi.it (L.F.); silvia.motta2@unimi.it (S.M.); carlo.santaniello@policlinico.mi.it (C.S.); monica.miozzo@unimi.it (M.M.); 2Medical Genetics, Department of Pathophysiology and Transplantation, Università degli Studi di Milano, 20122 Milan, Italy; silvia.tabano@unimi.it (S.T.); patrizia.colapietro@unimi.it (P.C.); 3Laboratory of Medical Genetics, Fondazione IRCCS Ca’ Granda Ospedale Maggiore Policlinico, 20122 Milan, Italy; 4Department of Brain and Behavioral Sciences, Università di Pavia, 27100 Pavia, Italy; luigi.corsaro01@universitadipavia.it; 5Medical Genetics, Department of Health Sciences, Università degli Studi di Milano, 20122 Milano, Italy; silvia.sirchia@unimi.it; 6Bioinformatics and Statistical Genomics Unit, Istituto Auxologico Italiano IRCCS, 20095 Cusano Milanino, Italy; luciano.calza@gmail.com; 7Department of Obstetrics and Gynecology, Fondazione IRCCS Ca’ Granda Ospedale Maggiore Policlinico, 20122 Milan, Italy; tatjana.radaelli@policlinico.mi.it (T.R.); enrico.ferrazzi@policlinico.mi.it (E.F.); 8Department of Clinical Sciences and Community Health, University of Milan, 20122 Milan, Italy; 9Scientific Direction, Fondazione IRCCS Ca’ Granda Ospedale Maggiore Policlinico, 20122 Milan, Italy; silvano.bosari@policlinico.mi.it

**Keywords:** placenta, methylome, LINE-1, birth weight, normal pregnancies

## Abstract

The placental methylation pattern is crucial for the regulation of genes involved in trophoblast invasion and placental development, both key events for fetal growth. We investigated LINE-1 methylation and methylome profiling using a methylation EPIC array and the targeted methylation sequencing of 154 normal, full-term pregnancies, stratified by birth weight percentiles. LINE-1 methylation showed evidence of a more pronounced hypomethylation in small neonates compared with normal and large for gestational age. Genome-wide methylation, performed in two subsets of pregnancies, showed very similar methylation profiles among cord blood samples while placentae from different pregnancies appeared very variable. A unique methylation profile emerged in each placenta, which could represent the sum of adjustments that the placenta made during the pregnancy to preserve the epigenetic homeostasis of the fetus. Investigations into the 1000 most variable sites between cord blood and the placenta showed that promoters and gene bodies that are hypermethylated in the placenta are associated with blood-specific functions, whereas those that are hypomethylated belong mainly to pathways involved in cancer. These features support the functional analogies between a placenta and cancer. Our results, which provide a comprehensive analysis of DNA methylation profiling in the human placenta, suggest that its peculiar dynamicity can be relevant for understanding placental plasticity in response to the environment.

## 1. Introduction

The placenta is the transient organ of primary importance during pregnancy, acting as a highly specialized interface intimately connecting mother and fetus. It ensures an adequate provision of nutrients and oxygen to the embryo as well as waste disposal and hormone production to support pregnancy and promote fetal growth. In order to allow the pregnancy to progress, even in presence of unfavorable conditions (of maternal or fetal origin), the placenta can adapt dynamically. Several factors, including in utero environmental changes and adverse maternal conditions such as infections, metabolic changes, and poor nutrition or obesity can induce placental adaptability through fine regulation processes that are not fully understood [1,2]. Despite the adaptability properties, early defects in the structure or function of the placenta or some maternal/fetal pathologic conditions can severely alter the maternal–fetal interface causing abortion, fetal death, abnormal fetal growth and maternal complications, including preeclampsia [3,4]. The causes of placental failure are often of genetic origin: mutated genes during conception, confined chromosomal unbalances, or genomic imprinting disturbances [5,6]. All these conditions can affect the gene expression equilibrium during pregnancy and, therefore, undermine the outcome.

Beyond the presence of recognizable genetic defects, gene expression modifications in crucial windows of placental development could affect its functioning, thus modifying maternal and fetal physiology. These placental changes can be difficult to identify for different reasons: accessibility to placental cells during pregnancy, morphological heterogeneity and the complex cellular composition of the organ. Despite the difficulties in recognizing the genetic/epigenetic alterations arising during pregnancy, they can represent biological markers of a pregnancy outcome.

Among the adverse factors, maternal undernutrition or poor nutrient intake due to a malfunction of placental transport can induce metabolic modifications in the fetus to maximize the limited nutrient resources and maintain survival. This process could be favored by epigenetic modifications to finely control the expression of genes crucial for fetal metabolic homeostasis. These signatures may persist after birth as an “epigenetic memory” of the metabolic adaptation during the prenatal life. A concept, in keeping with the DoHad hypothesis introduced by David Barker [7], links poor prenatal growth with susceptibility to metabolic disorders throughout life, such as type 2 diabetes and metabolic syndrome.

Epigenetics regulates gene expression without altering the DNA sequence, by combining the actions of enzymes that allow or inhibit gene expression [8]. It also functions as an interface between external inputs (e.g., drugs, infections, diseases, aging, diet, stress) and gene expression [9].

The development itself is an epigenetic process. This means that during fetal life the epigenome of the embryonic and extraembryonic compartments undergoes continuous change, which is essential for cell lineage differentiation, fetal and placental growth, and responses to external factors [10]. The pattern of placental methylation is crucial for the regulation of genes involved in trophoblast invasion and placental development, which are key events for fetal growth [11,12]. For example, the expression of the gene encoding chorionic gonadotropin is related to the gestational week of pregnancy and its expression is epigenetically regulated [13].

The DNA methylation landscape of the placenta is distinct from that of other tissues. The trophoblast and the mesenchyme core of the chorionic villi have a unique methylation profile compared with maternal decidua, fetal membranes and embryonic tissues [14]. The presence of large blocks (>100 kb) of intermediate methylation, named Partially Methylated Domains (PMDs), accounting for about 37% of the placental genome, can help explain the typical hypomethylation observed in the placenta, compared to other tissues [15,16].

In adult somatic tissues, DNA methylation is characterized by homogeneous distribution where most of the CpG sites show either high (> 90%) or low (<10%) methylation. In the placental DNA, however, the remarkable presence of PMDs results in a unique distribution of the methylation profile, characterized by peaks with partially methylated and highly methylated regions [16].

The hypomethylation of different families of Retrotransposable Elements (REs) is more pronounced in the placenta compared to other tissues, and the specific methylation pattern depends on the REs family and the evolutionary age of the sequence [17,18,19]. Furthermore, the methylation status of REs, such as the Long Interspersed Nuclear Elements (LINE-1), appears to be related to their location within or outside the PMD domains [16].

The placental methylome undergoes dynamic changes during gestation [18,20] not only to promote fetal development but also in response to environmental exposure that requires adaptive changes in the cellular composition of the placenta [20,21]. Fetal/placental adaptation to unfavorable situations, such as poor nutrient availability, can be achieved by epigenetic modification that can persist, thus modifying the physiological expression of genes in the fetus [22]. The dynamic of DNA methylation in fetal/placental compartments may be one of the mechanisms by which environmental exposure and genetic variation may influence fetal growth [23,24]. Several studies have shown that fetal growth and birth weight are significantly associated with global and specific methylation profiles of defined loci and CpG sites in the placental genome [25,26,27,28,29,30,31]. However, contradictory results are reported on the association of placental methylation level of LINE-1 and fetal growth defects [32,33,34].

A recent study, explored genome-wide DNA methylation in placental samples and identified 15 CpG sites with a methylation profile associated with birth weight [35]. These sites include loci already known to be associated with lipid metabolism in adults, inflammation and oxidative stress. The authors also reported methylation changes in genes associated with a lower birth weight in underweight mothers, preeclampsia and adult type 2 diabetes. Other studies described data on the methylation profile of four of these genes (*MLLT1*, *PDE9A*, *ASAP2*, and *SLC20A2*) in cord blood samples in relation to birth weight; on the other hand, increased methylation and lower expression of *FOSL1* correlated with a high birth weight [23,24,36,37,38]. Gene-related methylation changes associated with fetal growth in the human placenta have recently been reviewed by O’Callaghan et al. [34].

In this study we explored the methylome and LINE-1 methylation of embryonic (cord blood) and extraembryonic (placenta) tissues from 154 uncomplicated full-term pregnancies, stratified according to newborn weight percentile. The results were investigated by means of an ontology-driven approach and by comparing the results obtained by methylation microarrays and targeted sequencing.

## 2. Results

### 2.1. Study Design

We studied 154 pregnancies that were stratified according to birth weight percentile: 33 newborns were ranked as “small for gestational age” or SGA (≤10th percentile), 77 as “appropriate for gestational age” or AGA (>10th and <90th percentile), and 44 neonates as “large for gestational age” or LGA (≥90th percentile). This classification was based on the Italian infant growth charts developed by Bertino et al. [39]. Regarding the clinical features of the pregnancies, we found strong differences in maternal pregestational body mass index and placenta weight among SGA, AGA, and LGA groups (anova test, *p*-value < 0.001). The pregestational body mass index in the SGA group is on average 20.6 kg/m^2^, and in LGA group increases remarkably to 24.24 kg/m^2^. The placenta weight on average was 475 g in the SGA group and 741 g in the LGA group, respectively (Table 1).

An overview of experimental design is shown in Figure 1. We performed a LINE-1 methylation analysis in all pregnancies on maternal peripheral blood, cord blood and placental samples (462 DNA samples); a methylation array on cord blood and placenta samples from 10 pregnancies (20 DNA samples); and targeted methylation sequencing on cord blood and placenta samples from 5 other pregnancies (10 DNA samples). Concerning the consistency of the investigated population for methylation profiling by sequencing and array, Peng Liu et al. [40] recently published the MethylSeqDesign package, which is useful for inferring the Expected Discovery Rate (EDR: the number of claimed true positives/number of total true positives). After applying this tool, we found that a sample size of 10 samples per group is enough to obtain an EDR of 0.82. For sequencing, we had 5 samples per group, and based on the results published in Peng Liu et al. [40] concerning a pilot analysis with Agilent SureSelectXT Mouse Methyl-seq kit, we might exceed an EDR of 0.6.

For both methylation profiling array and targeted methylation sequencing approaches, samples were randomly selected across all pregnancies. We decided to perform these experiments on 15 pregnancies with different neonatal birth-weight (7 SGA, 7 LGA, and 1 AGA) in order to avoid selecting a specific birth-weight category and to explore possible evident differences between low and high birth weight classes.

### 2.2. LINE-1 Methylation Analysis

We performed the analysis on maternal and cord blood and placenta samples of 154 pregnancies (Figure 1).

As shown in Figure 2A, we found a significant LINE-1 hypomethylation in the placenta and compared it with both cord blood and maternal peripheral blood (The mean methylation ± standard deviation (%): 39.86 ± 6.08; 63.14 ± 4.88; 62.44 ± 4.47, respectively; p-value of placenta vs fetal blood: 4.5 × 10^−115^). Notably, methylation levels in cord and maternal blood were very similar, thus suggesting that LINE-1 methylation pattern in blood is stable and does not change throughout life.

Concerning placentae, by stratifying samples according to the birth weight percentile, we found a significant hypomethylation in the placental DNAs of SGA babies compared with AGA (*p* = 1.67 × 10^−4^) and LGA (*p* = 6.36 × 10^−5^) newborns. No differences were observed between AGA and LGA placentae (Figure 2B).

In addition, neither maternal or cord blood LINE-1 methylation differs among SGA, AGA, and LGA groups.

### 2.3. Methylation-Profiling Array

The analysis was carried on cord blood and placenta from 10 pregnancies (Figure 1) using the Infinium EPIC array approach.

The Principal Component Analysis was adopted to reduce the complexity of whole genome methylation data. The results, shown in Figure 3, clearly indicate evident differences between the two tissues: clustered distribution of cord blood samples (green circle) and scattered distribution of placental cases (orange triangle). Therefore, each placenta showed a unique methylation profile unrelated to the neonatal birth-weight percentile.

A differential analysis between placental and cord blood tissues was performed both at site and at the region level. After a false discovery rate (FDR) adjustment of the p-values for multiple testing, significant differences in methylation levels between cases and controls emerged at the site level. The list of probes that resulted in different methylation has been reported in the Appendix A. Differential methylation analysis performed at the regional level was carried out considering CpG islands, promoters, genes, and tiling. Results of the differential methylation analysis at the probe and regional levels were represented by volcano plots as shown in Figure 4.

### 2.4. Targeted Methylation Sequencing by NGS

The analysis was carried out on cord blood and placental DNA from five pregnancies (Figure 1) using SureSelect Human Methyl-Seq approach.

By PCA we found that the main source of variability in the methylation levels was between the two tissues, allowing us to distinguish specific CpG methylation clusters (Figure 5) and confirming data obtained by the methylation array approach.

Hierarchical clustering based on a CpG methylation score in cord blood and placenta (Figure 6) demonstrates that a general hypomethylation in placenta samples compared to cord blood was assessed, thus confirming data of LINE-1 methylation. In particular, Figure 6A depicts the distribution of each methylation value. The data show that the methylation profile of the placenta is not only composed of fully methylated or unmethylated regions but includes a wider range of regions displaying intermediate methylation scores compared to cord blood. This intermediate methylation status is likely due to PMDs, typical components of the placental genome [16].

In order to define the differentially methylated sites in cord blood and placenta, we selected the 1000 most variable sites and performed hierarchical clustering. We were thus able not only to distinguish between the two tissue clusters based on their opposite methylation patterns, but we also confirmed that the majority of differentially methylated sites are hypomethylated in the placenta (Figure 6B).

For the differential methylation module, we used the RnBeads rank, combining the mean methylation differences, methylation ratio, and *p*-value, to identify differences between the placenta and cord blood at the CpG site and at the region level, including tiling (5 kilo base pair regions), genes, promoters and CpG islands (Figure 7A).

We selected the 1000 most differentially methylated promoter regions and, to distinguish promoters based on their tissue-specific methylation status, stratified them into two subgroups. We identified 119 promoter regions methylated in the placenta and hypomethylated in the cord blood, and 881 promoters methylated only in the cord blood (Figure 7B).

By means of the Ingenuity Pathway Analysis (IPA) software v.60467501 (QIAGEN, Hilden, Germany), we explored which genes are differentially methylated in the promoter and in the gene body. We found that 75 promoters, specifically methylated in the placenta, are involved in blood cell proliferation and immune cell migration and adhesion, as expected (Table 2A). Differently, we found that promoter methylation was lower in the placenta than in the cord blood upstream of 450 genes that belong to pathways mainly involved in cancer (Table 2B and Appendix A).

Considering the results for gene body methylation of the 1000 most differentially methylated loci in placenta, we identified 53 genes specifically methylated in the placenta, 12 of which also displayed hypermethylation at their promoters in the same tissue. We found that they are mainly involved in blood cell-specific biological functions such as lymphocyte and leukocyte proliferation and migration. Among the 1000 genes, 404 were specifically methylated in cord blood, and 111 of them displayed hypomethylation in both the promoter and gene body in the placenta. The most represented pathways of the 404 genes are involved in several cancers, especially skin tumors and neuroendocrine cancers (Table 2B and Appendix A).

Considering the possible methylation differences in the birth weight groups, we used the RnBeads Differential Methylation module and investigated the differences in methylation among placental tissues. We did not find significant differences between SGA and LGA placentae, which could be explained by the small sample size as shown in Appendix A and Appendix A.

Finally, we focused our attention on *IGF2* and *H19* imprinted genes and master regulators of placental and fetal growth [41,42]. They were not among the top 1000 differentially methylated loci not in the promoters, or in the other investigated regions.

### 2.5. Comparison of the Methylation Profiles Obtained by Microarray and Targeted Sequencing

Although the comparison between two independent genome-wide techniques is biased because they consider different target regions, we tried to compare the data to identify regions with similar methylation status. Indeed, compared with an EPIC array, targeted methylation sequencing detects more CpG sites in coding regions and CpGs islands, whereas the proportion of CpG sites mapped on regulatory elements is comparable between the two techniques [43].

We applied the pipeline for differential methylation and pathway analyses to the datasets derived from both techniques. Analyzing the 1000 most differentially methylated loci in placenta compared to cord blood, we identified by microarray and methyl-seq, respectively: 209 and 450 hypomethylated promoters, 56 of which were in common between the two techniques; 340 and 75 hypermethylated promoters, 58 of which were common to the two techniques; 224 and 404 hypomethylated gene bodies, 74 of which were shared in the two approaches; 212 and 53 hypermethylated gene bodies, 35 of which in the two approaches gave similar results (see Appendix A).

Pathway analysis of these genomic loci returned similar results for data derived from microarray and targeted methylation sequencing, as both techniques identified an association with blood cell specific biological functions for loci that are hypermethylated in placenta, and an association with tumor-specific pathways for loci that are hypomethylated in the placenta (Table 3).

## 3. Discussion

The key role of the placenta in supporting pregnancy and promoting fetal growth is made possible by the capacity of this organ to modify its epigenetics during pregnancy in response to external factors.

This study explores the methylation signatures in the fetal and placental genome in uncomplicated full-term pregnancies with healthy newborns classified in three ranges of birth weight percentiles (SGA, AGA and LGA). As expected, we found that the maternal pregestational body mass index and placenta weight were strongly different in SGA, AGA and LGA groups. 

LINE-1 methylation analysis of the entire population (154 pregnancies) showed hypomethylation in placenta compared with both cord and maternal blood, confirming previous evidence [17,18,19]. Importantly, the LINE-1 methylation ranges in fetal and maternal blood were very similar to each other and independent from birth weight categories, suggesting that LINE-1 methylation deviations are possible in the placenta but not in other tissues (excluding cancers), which probably maintains a closer methylation pattern and, therefore, a stronger stability in these repetitive sequences throughout life. 

Less clear is a possible relationship of LINE-1 methylation and birth weight because of previous contradictory data [32,33,34], which are probably due, at least in part, to the non-homogeneous selection of cases. Herein, we found that LINE-1 methylation was significantly lower in the placentae of SGA babies compared with those of both AGA and LGA newborns. Given that, the hypomethylation of RE-derived promoters contributed to the placental-specific expression of genes or gene isoforms important for placental functions and development [44]. A more pronounced hypomethylation in the placentae of SGA infants could force the expression of these genes to support fetal growth. Alternatively, it is possible that LINE-1 hypomethylation could inhibit correct fetal growth in early development although no data are available in the literature to support this hypothesis.

To explore the placental methylome compared with cord blood in normal pregnancies and to define the main gene pathways of differentially methylated loci, we used a combined omics approach of methylation microarray and targeted sequencing by NGS.

The exploratory analysis clearly showed a significant difference between placenta and cord blood epigenetic profiles, a result not completely unexpected since the two tissues are substantially different. Another important result emerged from this analysis: a significant difference in the epigenetic variability between the two tissues. In particular, cord blood methylation profiles were very similar and therefore clustered, whereas the placental cases were variable and widespread. These findings show the plasticity of the placenta in relation to the uniqueness of each pregnancy. From our study, there emerged a unique methylation profile in each placenta compared to the cord blood, which could represent the sum of adjustments that the placenta makes throughout the course of the pregnancy to preserve the epigenetic homeostasis of the fetus, thus contributing to the delivery of a healthy baby. On the other hand, it is known that the placenta is more prone to sustain genetic alterations. For example, mosaic trisomies confined in the placenta are much more tolerated than in the fetus [45,46]. Similarly, placental plasticity was also evidenced in mice, even though the placentae in the two species display remarkable morphological variations. Decato et al. [47] conducted a study of placental DNA methylation landscape in multiple mouse strains and showed evidence of a dynamic methylation program associated with a highly heterogeneous and deregulated global landscape, which was also related with the developmental timepoints.

Methylation sequencing showed a global hypomethylation of placenta compared to cord blood, thus confirming our results on LINE-1 and those previously reported [15,16]. This hypomethylation is not restricted to a certain class of repeated elements but seems to be a feature of the entire placental genome (gene promoters, gene bodies, CpG islands and tilings).

The differential methylation analysis confirmed that DNA methylation profiles of cord blood and placenta are different both at site and region levels. This is clearly shown in the volcano plots. Interestingly, both the array and sequencing experiments achieved comparable results.

By restricting the analysis to the 1000 most variable sites between the two tissues, we could appreciate their opposite methylation patterns, thus highlighting the strong epigenetic differences of embryonic and extraembryonic compartments. We investigated the biological functions associated with these differentially methylated loci, by performing ontology pathway analysis on promoters and gene bodies. In particular, we found that promoters and gene bodies that are hypermethylated in the placenta and hypomethylated in cord blood are associated with blood cell specific functions such as proliferation and migration of immune cells, as would be expected for this tissue. However, promoters and gene bodies that are hypomethylated only in the placenta belong to six pathways involved in cancer and two related to ventricular myocytes.

In addition, we verified whether the methylation levels of *IGF2-* and *H19*-imprinted genes, the master loci of fetal and placental growth, were variables among samples. There were no differences either between cord blood and placenta or among the samples, thus reinforcing the concept that genomic imprinting, once established, remains stable [48].

In order to obtain relevant results we included only normal full-term pregnancies and found methylation NGS results in line with the EPIC array findings, despite the low number of cases. These analyses also highlighted significant overlapping differences between cord blood and the placenta at the most differentially methylated genome-wide sites. However, a point of weakness of our study is related to the small group size of neonates stratified for the birthweight analyzed with omics approaches; therefore, we could not find specific methylation-profiling related to birth weight. Nevertheless, a LINE-1 investigation carried out on the entire population revealed a peculiar hypomethylation associated with SGA phenotype.

As with the other authors [16], we showed that the placenta is characterized by regions displaying intermediate methylation scores likely due to PMDs, which are specific components of the placental genome. This can explain the hypomethylation that characterizes the placenta, the only normal tissue with PMDs. Only cancers share this peculiarity, and it has been hypothesized that PMDs may activate placental-specific programming in cancer cells [16]. The placental villi, similarly to those found in a cancer, are indeed able to migrate, invade, and remodel the maternal decidua. Moreover, as reported by Lorincz and Schubeler [49], the concept of similarity between epigenetic placental and cancer landscapes is particularly interesting: both adopt similar methylation adaptations to achieve analogous behaviours (e.g., rapid growth, vascular remodeling, and cell invasion). These common behaviors may explain why activated pathways are shared between these tissues. Accordingly, our results showed that the most relevant Functions Annotation of pathways active in placenta were related to different types of cancer including skin and neuroendocrine tumors.

Taken together, our findings confirmed the global hypomethylation of placental DNA including LINE-1, promoters, CpG islands, gene body, and tilings. They also showed that specifically activated pathways in the placenta are those involved in cancer. The placental heterogeneity of the methylation levels characterizing each pregnancy could explain its broader plasticity compared to the fetal compartment.

## 4. Materials and Methods

### 4.1. Study Population

Recruitment was carried out at the Unit of Obstetrics and Gynecology “L. Mangiagalli” (Fondazione IRCCS Ca’ Granda Ospedale Maggiore Policlinico, Milan, Italy). Participants were enrolled before the delivery either by cesarean section or vaginal birth. Inclusion criteria were: European origin of the mothers, singleton spontaneous pregnancies delivered full-term, spontaneous conception, absence of fetal malformations or known genetic abnormalities, and the absence of severe pregnancy complications (including intra-uterine growth restriction, hypertensive disorders or preeclampsia). For each pregnancy, a medical history, clinical data and maternal and neonatal outcomes were recorded. The study protocol was approved by the local Ethical Committee (reference ID number 2487-588ter, 28.04.2015), and written informed consent was obtained from all participants. This study was part of a broader project aimed at collecting 2000 normal pregnancies to create a biorepository of maternal and fetal tissues for present and future studies.

### 4.2. Sample Collection and DNA Extraction

For each pregnancy at delivery, we collected maternal peripheral blood and umbilical cord blood in EDTA tubes (ThermoFisher, Waltham, MA, USA) and 1 cm³ of placental tissue taken from the fetal side in a cotyledon closed to cord insertion. All samples were frozen at −80 °C within 4 h of sampling and transferred in the repository Biobank of Fondazione IRCCS Ca’ Granda Ospedale Maggiore Policlinico.

DNA was extracted from the cord blood, maternal peripheral blood and placenta tissue using the QIAsymphony (QIAGEN, Hilden, Germany) automated DNA extraction platform, according to the manufacturer’s instructions. DNA quality and concentration was assessed by NanoDrop and Qubit (ThermoFisher, Waltham, MA USA) analysis. DNA integrity and fragment size were assessed by TapeStation analysis (Agilent, Santa Clara, CA USA). DNA bisulfite conversion was performed using the EZ DNA Methylation Gold Kit (Zymo Research, Irvine, CA, USA) for LINE-1 and methylation sequencing experiments, and the EZ DNA Methylation Kit (Zymo Research, Irvine, CA, USA) for the microarray.

### 4.3. Methylation Profiling

#### 4.3.1. LINE-1 Methylation Analysis

LINE-1 methylation was carried out on the placental, maternal and cord blood DNA of all cases (Figure 1) in order to measure the global LINE-1 methylation pattern in the placenta to compare it with adult and cord blood, and to assess possible relationships with the birth-weight percentile.

LINE-1 methylation analyses were performed as previously described [41]. In brief, the PyroMark Q96 CpG LINE-1 kit (QIAGEN, Hilden, Germany) was used to amplify LINE-1 sequences from 20 ng of bisulfite-converted DNA and to quantify methylation levels at four CpG sites. Quantitative DNA methylation analyses were performed using pyrosequencing, using the Pyro Mark ID instrument (QIAGEN, Hilden, Germany), equipped with PSQ HS 96 System, and PyroGold SQA reagent kit (QIAGEN, Hilden, Germany), according to the manufacturer’s instructions. Raw data were analyzed using the Q-CpG software v1.0.9 (Biotage AB, Sweden), which calculates the ratio of converted C’s (T’s) to unconverted C’s at each CpG, giving the percentage of methylation [50]. Reported methylation values are the mean between at least two independent PCR and pyrosequencing experiments.

#### 4.3.2. Methylation-Profiling Array

##### Genome-Wide Methylation Analysis

Genome wide methylation was carried out using the Infinium MethylationEPIC array (Illumina, San Diego, CA, USA) on both cord blood and placental DNAs from ten pregnancies (5 SGA and 5 LGA) (Figure 1) in order to define the methylation profiles in these embryonic tissues. Infinium MethylationEPIC array interrogates over 850,000 sites across the genome at single-nucleotide resolution.

To evaluate the conversion efficiency and bsDNA integrity, a single-strand quantification of bsDNA was performed using NanoPhotometer Pearl (Implen GmbH, Germany).

##### Data Management, Pre-Processing, Normalization, and Quality Control

We used the RnBeads package (2.8.0) in the R environment (Version 4.0.0) and adopted all the available modules to carry out the quality control, normalization, exploratory analysis (with Principal Component Analysis) and differential analysis of all the genomic regions (genes, promoters, CpG islands and tiling).

In the preprocessing phase 826,020 Sites of SNP-enriched probes were counted; 17,371 sites were removed because they were unreliable; and 2985 sites were context-specific. In addition, all the sex chromosome probes were removed (19,438 sites).

After the filtering procedures, 849,524 CpG sites were retained and all samples met the minimum criteria for being included in the study.

The probes methylation values were normalized using the MINFI (1.36.0) package of the SWAN normalization method [51] and by the limma method. We computed the p-values of grouped samples in order to compare differentially the methylation levels of the two groups.

A combined p-value of the aggregated probes by regions (genes, promoters, CpG island, tiling) was calculated and used for the differential methylation analysis.

The outcome for the four regions was carried out as follows: (i) genes, represented with the Ensembl format version 75 (*n* = 33,842); (ii) Promoter regions of Ensembl genes, version Ensembl Genes 75 (*n* = 43,436); (iii) CpG Islands, CpG island track of the UCSC Genome browser (*n* = 25,769); and iv) Tiling regions, non-overlapping tiling regions with a fixed window size of 5 kilobases defined over the whole genome (*n* = 246,489).

#### 4.3.3. Targeted Methylation Sequencing

##### Methylation-Capture Sequencing

For a complete genome-wide methylation analysis, we performed Methylation Sequencing in cord blood and placental DNA from five pregnancies (2 SGA, 2 LGA and 1 AGA) (Figure 1) using the SureSelect Human Methyl-Seq (Agilent, Santa Clara, CA USA), a target enrichment protocol based on bisulfite-conversion which detects DNA methylation at single nucleotide resolution and allows the analysis of over 3.7 million individual CpG dinucleotides which fall into under- and over-methylated cytosine sites in the human genome. Libraries preparation was performed using the SureSelect XT Methyl-Seq Target Enrichment kit (Agilent, Santa Clara, CA USA), sequenced using 100 bp paired-end and run on the Illumina NextSeq550 (Illumina, San Diego, CA, USA).

##### Data Management, Pre-Preprocessing, Normalisation and Analysis

Base calls and relative quality scores were computed from signal intensities during the sequencing run by the system’s Real Time Analysis (RTA) software. Sample de-multiplexing was performed by the Illumina bcl2fastq Conversion Software v2.2 resulting in sequence data in FASTQ format. Files corresponding to each of the four NextSeq550 lanes were merged with Samtools-merge.

Subsequent steps were performed using the nf-core/methylseq v1.6dev Bismark (Babraham Bioinformatics, Cambridge, United Kingdom) workflow. First, quality control (QC) was performed by FastQC. The trimming step was skipped. Then, Bismark-aligner was used to align paired-end reads to bisulfite-converted versions of the human genome (hg19), while Bismark-deduplicate estimated and removed the duplicate reads. Finally, methylation calls in CpG, CHG and CHH context were extracted by a Bismark-methylation_extractor.

The resulting methylation data from Bismark was imported in the RnBeads v2.0.0 software for assay quality validation, CpG site filtering, sample clustering, and differential methylation analyses. During the RnBeads filtering step, CpG sites located in chromosomes other than 1–22 (including mitochondrial and sex chromosomes) were removed, along with CpG sites mapping at annotated single nucleotide polymorphisms. Only sites with at least 5× coverage and no missing values in all analysed samples were conserved. We obtained 2,871,130 CpG sites with available methylation data across all 10 samples. This allowed us to drastically decrease the proportion of unreliable measurements (0% as estimated by RnBeads) while still maintaining a sufficient amount of reliable measurements (over 75%). During the Exploratory Analysis step, RnBeads performed hierarchical clustering of samples and Principal Component Analysis (PCA) based on methylation scores of the filtered CpG sites. For the Differential Methylation analysis, *p*-values on the site level were computed using the limma method and adjusted with False Discovery Rate (FDR). RnBeads also computes a combined rank, taking into consideration not only significance from p-values, but also the mean of the methylation differences observed for that site in the two conditions being compared as well as the mean methylation ratio. Methylation at region level (for genes, promoters and CpG islands) is computed as the mean of methylation values of sites falling into the region, while the combined *p*-value is obtained by the weighted inverse chi-square method.

## Figures and Tables

**Figure 1 ijms-22-02136-f001:**
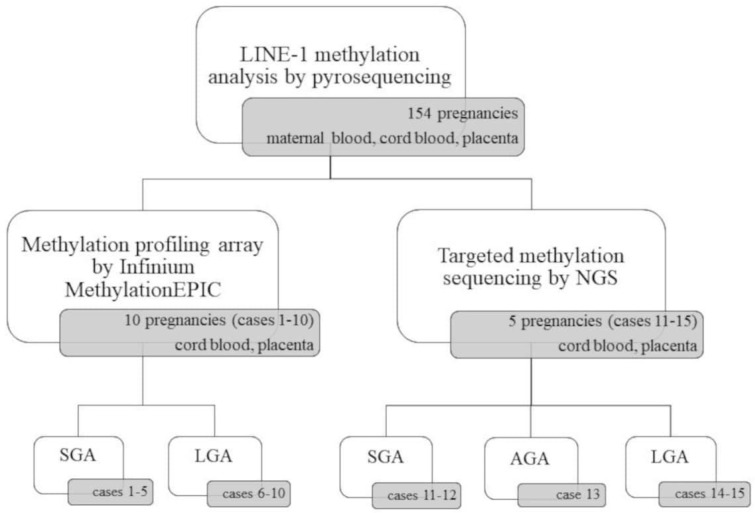
Schematic overview of the experimental design. SGA, small for gestational age; AGA, appropriate for gestational age; LGA, large for gestational age.

**Figure 2 ijms-22-02136-f002:**
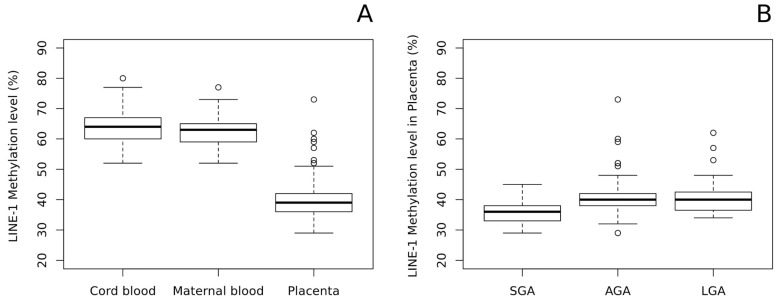
Box-plots of LINE-1 methylation results in 154 pregnancies. (**A**) Cord blood, maternal blood, and placental methylation level (%) distributions; (**B**) Placental methylation level (%) distributions stratified according to birth weight percentiles. The rectangles indicate lower, upper quartiles and the median methylation values. Data falling outside the whiskers are plotted as outliers of the data.

**Figure 3 ijms-22-02136-f003:**
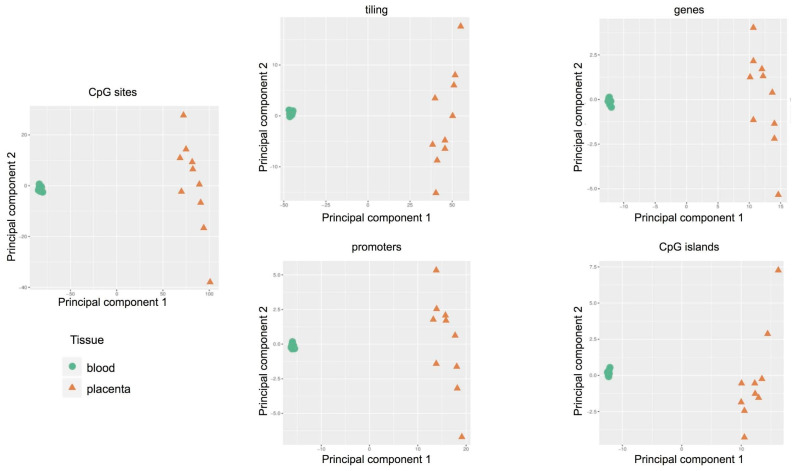
Methylation-profiling microarray: Principal Component Analysis (PCA) in 10 pregnancies (cases 1–10). Scatter plot showing the samples coordinates on principal components based on methylation scores of CpG sites and regions: tiling, genes, promoters, and CpG islands.

**Figure 4 ijms-22-02136-f004:**
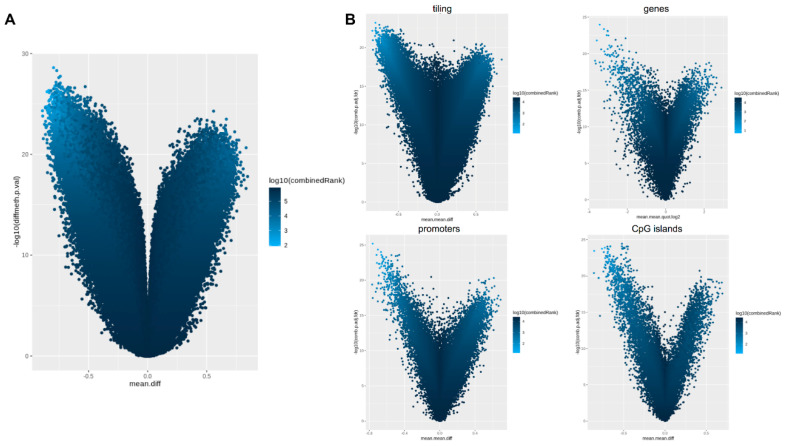
Methylation-profiling array: Volcano plot for differential methylation in 10 pregnancies (cases 1–10). (**A**) Volcano plot for differential methylation in CpG sites, quantified using methylation score differences between cord blood and placenta (*x*-axis) and the p-value of each region (*y*-axis). Points identify differentially methylated sites that are hypermethylated in placenta (left portion of the plots) or hypermethylated in cord blood (right portion of the plots). Color scale is based on a combined ranking. (**B**) Volcano plots for differential methylation in tiling, genes, promoters, and CpG islands.

**Figure 5 ijms-22-02136-f005:**
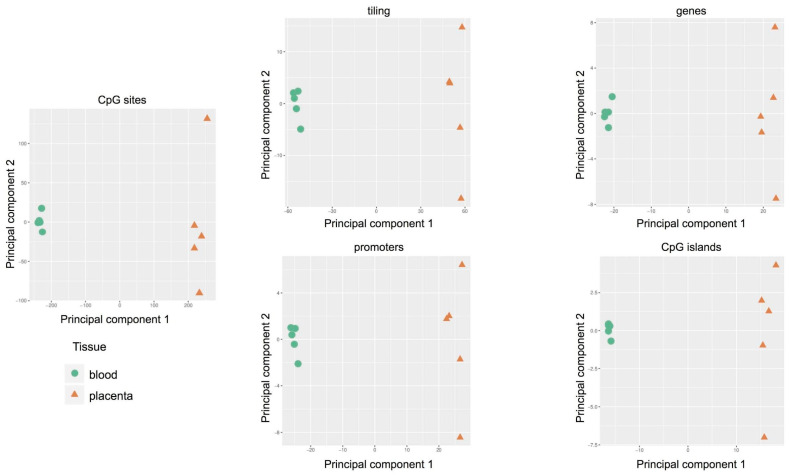
Targeted methylation sequencing: Principal Component Analysis (PCA) in five pregnancies (cases 11–15). Scatter plot showing the samples’ coordinates on principal components based on methylation scores of CpG sites (left) and regions (right): tiling, genes, promoters, and CpG islands.

**Figure 6 ijms-22-02136-f006:**
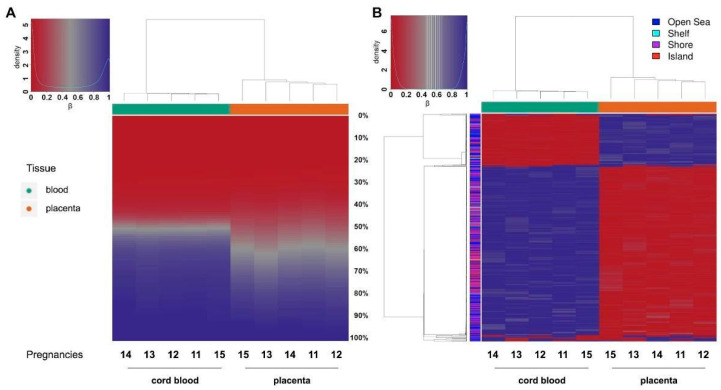
Targeted methylation sequencing: Hierarchical clustering based on CpG methylation score in cord blood (green) and placenta (orange). The heatmaps show (**A**) the relative distribution of CpG site methylation scores ranging from 0 (red) to 100% (blue) in the placenta (green) compared to cord blood (orange); (**B**) the methylation scores for the top 1000 CpG sites based on variance across samples. Color bar on the left indicates CGI relation for each site (shore: <2 kb flanking CpG island; shelf: <2 kb flanking outwards from CpG shore).

**Figure 7 ijms-22-02136-f007:**
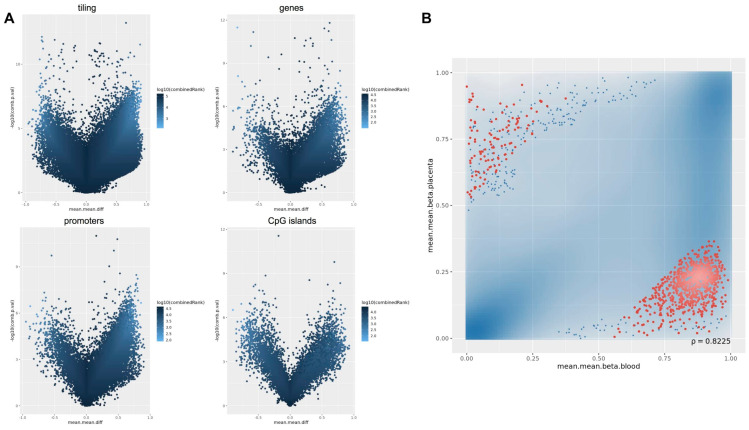
Targeted methylation sequencing: RnBeads differential methylation analysis. (**A**) Volcano plot for differential methylation in tiling, genes, promoters and CpG islands, quantified by the mean methylation score difference between cord blood and placenta (*x*-axis) and the combined adjusted p-value of each region (*y*-axis). Points identify differentially methylated regions that are hypermethylated in placenta (left portion of the plots) or hypermethylated in cord blood (right portion of the plots). Color scale is based on combined ranking, which we used to select the top 1000 differentially methylated regions (threshold: log_10_ (combined Rank) ≤3.312389); (**B**) Scatter plot showing differential methylation in promoters between cord blood (*x*-axis) and placenta (*y*-axis). The top 1000 differentially methylated promoters, based on combined ranking from p-value, methylation difference and methylation ratio, are highlighted in red. Blue shade clouds correspond to high point density, while the 1% of the points in the sparsest populated plot regions are drawn explicitly.

**Table 1 ijms-22-02136-t001:** Study population with the main clinical features of the pregnancies.

Weight Percentile of Neonates	SGA (*n* = 33)	AGA (*n* = 77)	LGA (*n* = 44)
Maternal age (y) mean ± SD	35.06 ± 4.53	34.79 ± 4.32	35.45 ± 3.68
Maternal pregestational weight (kg) mean ± SD	53.68 ± 6.80	59.71 ± 10.37	68.32 ± 16.61
Maternal pregestational body mass index (kg/m^2^) mean ± SD	20.06 ± 2.21	21.85 ± 3.71	24.24 ± 5.74
Gestational weight gain (kg) mean ± SD	9.05 ± 3.25	11.17 ± 3.87	11.67 ± 4.48
Gestational age at delivery (WOG) mean ± SD	38 ± 2	39 ± 2	39 ± 2
Mode of delivery (Vaginal Birth/ Cesarean Section) count; percentage	VB (n = 13; 8.44%) CS (n = 20; 12.99%)	VB (n = 24; 15.60%) CS (n = 53; 34.42%)	VB (n = 18; 11.69%) CS (n = 26; 16.88%)
Birthweight (g) mean ± SD	2486.67 ± 335.61	3360.91 ± 434.23	4109.77 ± 258.19
Placental weight (g) mean ± SD	475.19 ± 98.97	587.95 ± 97.12	741.47 ± 145.38

**Table 2 ijms-22-02136-t002:** Diseases or Functions Annotation from IPA Core Analysis on given gene datasets. The top 10 most significant annotations are shown.

A	
Diseases or Functions Annotation	*p*-Value
Leukocyte migration	1.52 × 10^−16^
Cell movement of lymphocytes	5.05 × 10^−14^
Cell movement of mononuclear leukocytes	6.30 × 10^−14^
Quantity of leukocytes	7.50 × 10^−14^
Lymphocyte migration	9.59 × 10^−14^
Quantity of lymphatic system cells	1.97 × 10^−13^
Quantity of lymphocytes	5.29 × 10^−13^
Cell movement of leukocytes	6.80 × 10^−13^
Proliferation of blood cells	1.21 × 10^−12^
Proliferation of immune cells	1.62 × 10^−12^
**B**	
**Diseases or Functions Annotation**	***p*-Value**
Cutaneous melanoma	1.52 × 10^−12^
Skin tumor	5.94 × 10^−12^
Melanoma	1.01 × 10^−11^
Skin cancer	1.20 × 10^−11^
Malignant neuroendocrine neoplasm	2.45 × 10^−7^
Small cell lung carcinoma	3.07 × 10^−7^
Olfactory response	3.73 × 10^−7^
Neuroendocrine tumor	4.48 × 10^−7^
Extrapancreatic neuroendocrine tumor	5.84 × 10^−7^
Blue round small cell tumor	7.16 × 10^−6^

(**A**) Functions identified from gene promoters methylated in the placenta but not in cord blood. (**B**) Functions identified from gene promoters methylated in cord blood but not in the placenta.

**Table 3 ijms-22-02136-t003:** Diseases or Functions Annotations in common between methylation-profiling microarray data and targeted methylation sequencing data (data from IPA Core Analysis).

A		
Diseases or Functions Annotation	*p*-Value (Microarray)	*p*-Value (Methyl-Seq)
Leukocyte migration	2.93 × 10^−26^	4.53 × 10^−7^
Quantity of lymphatic system cells	3.30 × 10^−25^	4.34 × 10^−6^
Quantity of leukocytes	1.51 × 10^−25^	3.42 × 10^−6^
Quantity of lymphocytes	1.53 × 10^−24^	4.75 × 10^−6^
Cell movement of leukocytes	5.69 × 10^−23^	1.38 × 10^−10^
Activation of cells	2.86 × 10^−22^	5.05 × 10^−14^
Migration of cells	1.08 × 10^−21^	7.35 × 10^−7^
Proliferation of lymphatic system cells	1.68 × 10^−21^	1.62 × 10^−6^
Proliferation of lymphocytes	5.96 × 10^−21^	2.09 × 10^−6^
Proliferation of immune cells	3.46 × 10^−20^	1.58 × 10^−6^
**B**		
**Diseases or Functions Annotation**	** *p* ** **-Value (Microarray)**	** *p* ** **-Value (Methyl-Seq)**
Cross-linkage of protein	1.60 × 10^−4^	8.28 × 10^−4^
Skin carcinoma	7.37 × 10^−4^	3.95 × 10^−3^
Skin squamous cell carcinoma	9.61 × 10^−4^	2.69 × 10^−4^
Relaxation of heart ventricle	1.03 × 10^−3^	5.24 × 10^−3^
Surface area of ventricular myocytes	1.22 × 10^−3^	6.15 × 10^−3^
Gastro-esophageal carcinoma	3.77 × 10^−3^	2.78 × 10^−3^
Chemosensitivity of glioblastoma cells	4.03 × 10^−3^	9.20 × 10^−3^
Pancreatic lesion	5.88 × 10^−3^	6.34 × 10^−3^
Formation of solid tumor	5.90 × 10^−3^	1.74 × 10^−3^

(**A**) Top 10 functions identified from gene promoters methylated in placenta but not in cord blood. (**B**) Functions identified from gene promoters methylated in cord blood but not in placenta. (Only 9 common annotations were found).

## Data Availability

The data presented in this study are available in the article’s supplementary material.

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
