# Peer review of "Extensive Placental Methylation Profiling in Normal Pregnancies"

_ijms, 2021, doi:10.3390/ijms22042136_

Round 1

Reviewer 1 Report

An interesting study demonstrating that the placenta is hypomethylated relative to adult maternal blood or cord blood and that methylation may differ in SGA pregnancies.

Major Comments:

  • Only 10 placentas are used for the methylation array and only 5 placentas are used for the targeted sequencing. Does this provide enough power? What was the reason for not using all 154 pregnancies?
  • Consider discussing the differences between maternal characteristics in your groups and the limited n number in the exploratory analysis in your discussion

Minor Comments:

  • Very minor English/grammatical errors throughout
  • Figure 3 and figure 5 need larger icons for visibility.

Author Response

Major Comments:

  • Only 10 placentas are used for the methylation array and only 5 placentas are used for the targeted sequencing. Does this provide enough power? What was the reason for not using all 154 pregnancies?

Thank you for this observation. The aim of the study was to carry out a pilot study on a small number of samples implementing different techniques. Despite a small sample size, the strength of achieved data is attested by the comparable results obtained by EPIC array and targeted methylation sequencing. In addition, for MethylSeq, only one study has previously investigated the power calculation, finding that a sample size of 5 samples per group is sufficient to reach an expected discovery rate (EDR) of 0.62. However, a sample size of 10 samples per group would be optimal to achieve an EDR of 0.8, as we discussed in Results 2.1 (lines 148-155).

  • Consider discussing the differences between maternal characteristics in your groups and the limited n number in the exploratory analysis in your discussion

Thank you for this suggestion: we further investigated the distribution of parameters reported on Table 3 and added this on the results chapter. We found a relationship between maternal pre-pregnancy body mass index and birth weight percentile, and an association between placental weight and birth weight percentile as discussed in Results 2.1 (lines 137-143). In addition, we included a discussion on the weaknesses of the study related to the small sample size (lines 389-397).

Minor Comments:

  • Very minor English/grammatical errors throughout
  • Figure 3 and figure 5 need larger icons for visibility. 

Figures 3 and 5 were modified as suggested.

Reviewer 2 Report

The authors carried out a substantial analysis about the DNA methylation profiling in human placenta. The study is very specific and the authors concentrated on LINE-1 methylation (in maternal blood, cord blood and placenta) together with methylation profiling array and targeted methylation sequencing (in cord blood and placenta). The results of the study appear interesting but there are some questions:

  1. Line 179: The authors stated that “a strong differences among samples from different pregnancies was evident”. Please, explain this sentence; in particular if there are differences by stratifying samples according to the birth weight percentile (SGA and LGA).
  2. Concerning the targeted methylation sequencing by NGS (paragraph 2.4), it isn’t clear in the Results section if there are any differences in the pattern of methylation considered between the three classes of pregnancies analysed (SGA, AGA, LGA). Even if the sample size of this group of study is too small (5 pregnancies, 2 cases for SGA, 1 for AGA and 2 for LGA as illustrated in Figure 1) it would be important to elucidate (in the Results section) and discuss this point (in Discussion section).
  3. It would be helpful to discuss the biological significance of the data, in particular the functional analogies of the results between placenta and cancer, as stated in the Discussion section (lines 367-369) together with an explanation of strengths and the weaknesses of the study.

Author Response

  1. Line 179: The authors stated that “a strong differences among samples from different pregnancies was evident”. Please, explain this sentence; in particular if there are differences by stratifying samples according to the birth weight percentile (SGA and LGA). 

We provided a clearer explanation of this sentence in Results 2.3 (lines 189-192); we could not appreciate any differences by stratifying samples according to the birth weight percentile.

  1. Concerning the targeted methylation sequencing by NGS (paragraph 2.4), it isn’t clear in the Results section if there are any differences in the pattern of methylation considered between the three classes of pregnancies analysed (SGA, AGA, LGA). Even if the sample size of this group of study is too small (5 pregnancies, 2 cases for SGA, 1 for AGA and 2 for LGA as illustrated in Figure 1) it would be important to elucidate (in the Results section) and discuss this point (in Discussion section).  

Thank you for the suggestion. We did not find differences in the methylation profiling using array and methylation approaches considering the three classes of pregnancies. However, we added the results on the comparison of methylation profiles considering the birth weight percentile in paragraph 2.4 and in Supplementary Figure 1 and Supplementary Tables 19-23 (Results 2.4, lines 285-287). In addition, in the Discussion we mentioned these results (lines 393-395) and in paragraph 2.1 we included a sentence about the consistency of the investigated population for methylome analyses (lines 148-155).

  1. It would be helpful to discuss the biological significance of the data, in particular the functional analogies of the results between placenta and cancer, as stated in the Discussion section (lines 367-369) together with an explanation of strengths and the weaknesses of the study.

As suggested, we enriched the discussion regarding: the biological significance of the data (lines 360-365); the functional analogies between placenta and cancer (lines 404-407); and an explanation of strengths and the weaknesses of the study (lines 389-397).